# DRIVE: Best Data Scheduling Practices for Reinforcement Learning wIth VErifiable Reward in Competitive Code Generation

**Speed Zhu** [1]   **Chuheng Zhang** [2]   **Jianwei Cai** [1]   **Guang Chen** [1]   **Lulu Wu** [1]   **Xiaolong Xu** [3]   **Xuyun Zhang** [4]
**Saiyong Yang** [1]   **Wiggin Zhou** [1]

## Abstract

Recent success of large reasoning models (such as OpenAI o1 and DeepSeek R1) have spurred a resurgence of interest in reinforcement learning from verifiable rewards (RLVR). However, progress is still largely driven by RL algorithm design, while data scheduling – the data-side decisions that determine what the model trains on over time – is critical but remains underexplored. Therefore, data scheduling becomes the focus of this paper, including how to curate data for supervised fine-tuning (SFT) and how to select prompts and collect rollouts for reinforcement learning (RL). We introduce a pipeline with careful designs on data scheduling, consisting of hardness-prioritized SFT and two-stage RL. Specifically, we first fine-tune the base model on supervision data that is curated to prioritize difficulty based on both arena learning and classification. Then, we introduce two-stage RL where a decreased max sequence length during rollout is used in the first stage to expand entropy and reduce repetition, and a large number rollouts per prompt and curriculum design are adopted in the second stage to encourage exploration for challenging problems. We implement this pipeline on Qwen2.5-32B and an internal 406B MoE model, and evaluate them on a wide range of benchmarks including challenging LeetCode and Codeforces weekly contests. The results not only indicate the effectiveness and scalability of our pipeline but also demonstrate our model achieve sota of 32B models in competitive code generation.

## 1. Introduction

Recent advances in large reasoning models (LRMs) (Besta et al., 2025), such as OpenAI o1 (Jaech et al., 2024), o3 (El-Kishky et al., 2025), DeepSeek R1 (Guo et al., 2025), and QwQ (Qwen Team, 2024), have demonstrated remarkable capabilities in complex problem-solving tasks through reinforcement learning with verifiable rewards (RLVR). RLVR optimizes the model via reinforcement learning (RL) based on the rewards that can be computed automatically. While math becomes a popular domain where such rewards can be easily obtained by matching the output with the ground-truth solution (Shao et al., 2024; Hu et al., 2025; Zheng et al., 2025; Li et al., 2025c), unit test in sandbox makes RLVR also viable for competitive code generation (Zhao et al., 2024; Zhang et al., 2024). Compared with math, competitive code generation presents unique challenges: 1) The solution should not only be correct in logic but also executable and efficient to yield answers within strict computational constraints, and 2) the code should be capable of handling different edge cases instead of just one problem instance as in typical math benchmarks such as AIME.

Prior work on RLVR has predominantly focused on developing novel algorithms and training techniques (see e.g., Shao et al., 2024; Yu et al., 2025; Cheng et al., 2025; Cui et al., 2025), with considerably less attention paid to the critical aspects of data scheduling, including how to curate data in SFT and how to rollout in RL. Existing approaches either train on a uniform distribution of the samples across different difficulties or rely on simple difficulty filtration (Seed et al., 2025; Yu et al., 2025; Li et al., 2025b; 2023), limiting the model's ability to tackle the most challenging problems.

In this work, we present a comprehensive solution to apply RLVR to competitive programming code generation, with particular emphasis on practical data scheduling, including data curation and rollout curriculum designs. In supervised fine-tuning (SFT), we train model based on the data that is curated to prioritize difficulty based on both arena learning and classification, to improve the performance on challenging problems. Specifically, we first use the arena learning strategy adapted from Luo et al. (2024) to select the hard samples, and then train a classifier to categorize the diffi-

---

[1]Hunyuan Team, Tencent [2]Independent Researcher [3]Nanjing University of Information Science and Technology [4]Macquarie University. Correspondence to: WigginZhou <wigginzhou@tecent.com>.

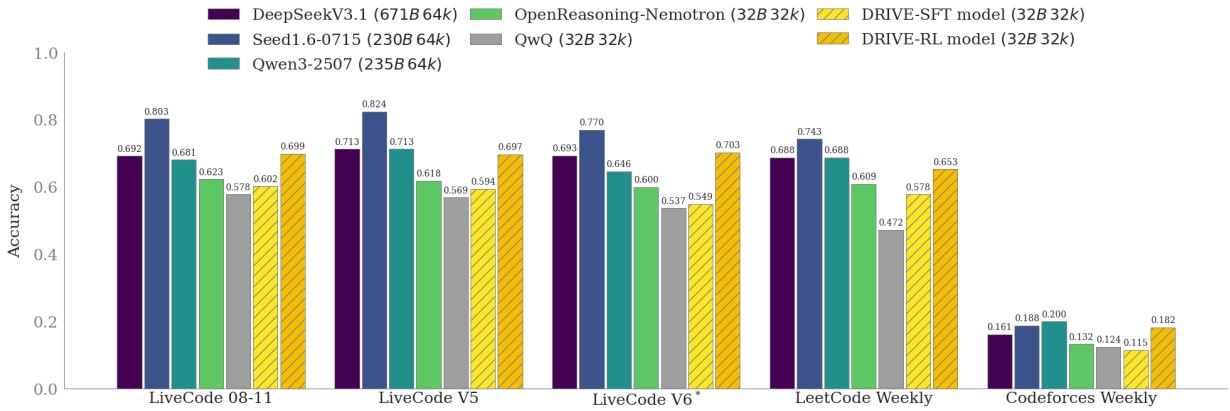

*Figure 1.* The performance of our pipeline and other baselines on various benchmarks. The numbers in the parentheses indicate the number of parameters and the max sequence length set during inference of different models.

culty of training samples and duplicate the hard samples. In reinforcement learning (RL), we adopt a two-stage scheme. In the first stage, we limit the sequence length of rollouts to encourage the model to generate shorter, less repetitive, but more diverse responses. We call this phenomenon *entropy expansion*, which is essential for efficient exploration in the next stage. In the second stage, we collect a large number of rollouts per prompt to encourage sufficient exploration, and thus improve the performance on hard instances.

We implement our method on Qwen2.5-32B (Hui et al., 2024) and demonstrate its effectiveness through extensive evaluation on recent LeetCode and Codeforces weekly contests, benchmarks carefully selected to avoid data contamination. We present the overall performance comparison in Figure 1. Our model achieves state-of-the-art performance among the models of similar size and remains competitive against much larger model such as DeepSeek V3.1 (Taşyürek et al., 2025), with particularly strong gains on the most challenging problems – up to 58% relative improvement over the models of similar size on Codeforces. Our comprehensive ablation studies show that entropy expansion results in robust generalization and the curriculum design extends the model's problem-solving frontier. In addition, we also observe that our pipeline can scale effectively to large-scale MoE models.

Our key contributions are:

- **A pipeline consisting hardness-prioritized SFT and two-stage RL**, with well-designed data scheduling in RLVR, that enables the model to excel at solving the most challenging coding problems.

- **The empirical evidence** demonstrating that 1) the small max sequence length can lead to entropy expansion and therefore encourage further exploration, and 2) the large number of rollouts per prompt is crucial for learning on challenging problems.

- **The state-of-the-art results** on competitive programming benchmarks among 32B models, even achieving performance comparable to models with $5\times$–$10\times$ more parameters. Moreover, we also achieve good performance on a large-scale MoE model.

In summary, our findings not only highlight the importance of weighing the hard samples more in competitive code generation but also suggest that careful curriculum design and strategic use of computational resources (e.g., setting max sequence length and the number of rollouts per prompt) can yield substantial improvements in competitive programming capabilities, providing insight for future work to tackle challenging problems.

## 2. Related Work

**RLVR Algorithms.** Since DeepSeek R1 (Guo et al., 2025) demonstrated successful RLVR implementation on both Qwen-32B and DeepSeekV3, numerous studies have investigated performance improvements, particularly on the AIME benchmark. DAPO (Yu et al., 2025) was the first open-source method to enhance RLVR performance on AIME using the GRPO algorithm with Qwen2.5-32B. Subsequently, VAPO (Yue et al., 2025b) proposed an enhanced PPO algorithm for further gains, while ProRL (Liu et al., 2025) explored GRPO optimization tricks to boost benchmark performance. Other works have focused on stabilizing RLVR training, including expert replay strategies for MoE models (Zheng et al., 2025) and truncated importance sampling to address mismatches between inference and training engines (Yao et al., 2025). However, despite these algorithmic advances, remarkably little attention has been paid to how the data are curated in SFT and rolled out – a gap that our work addresses.

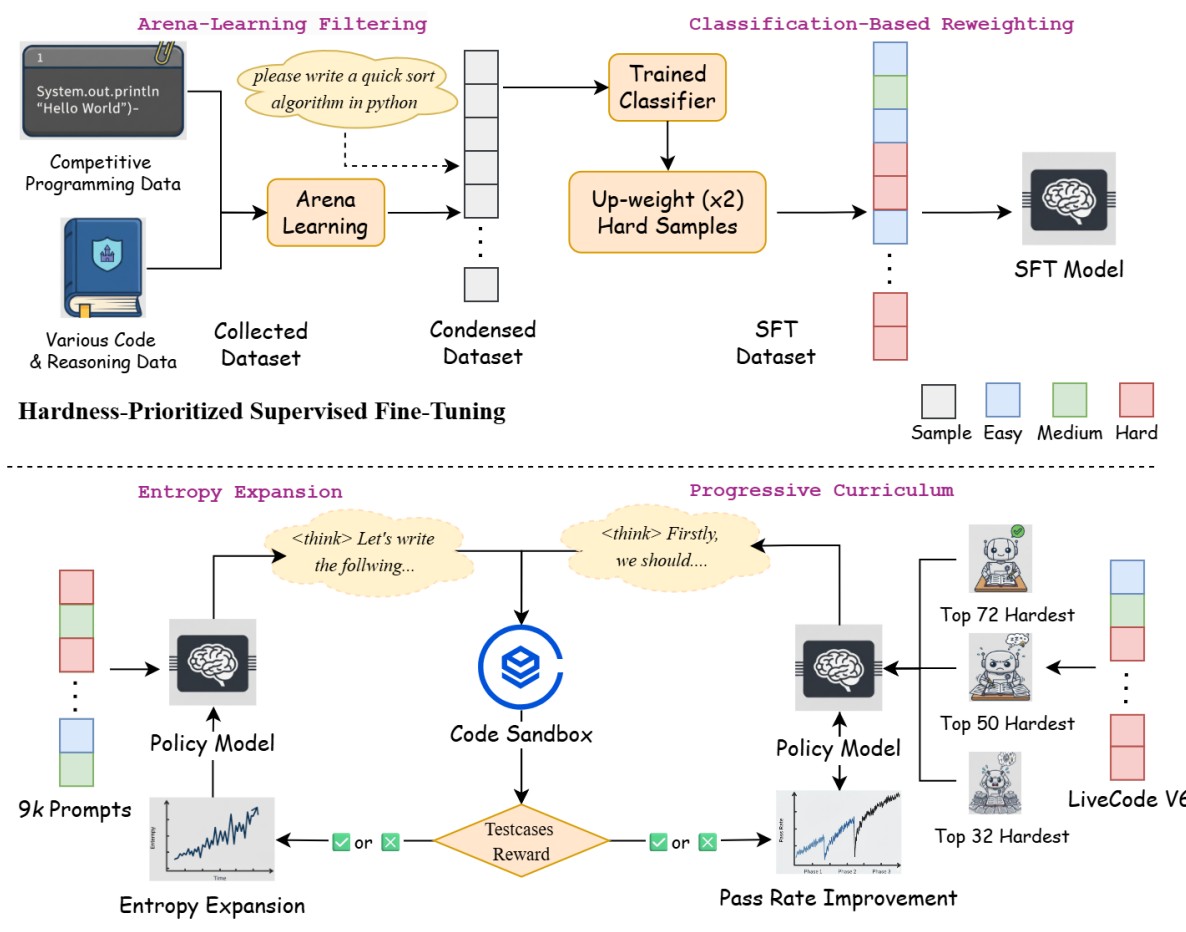

*Figure 2.* Our training pipeline consists of hardness-prioritized SFT and two-stage RL.

Closely related studies include Yue et al. (2025a) that points out that data curation should be a critical path to improve reasoning. They also suggest distillation to be a promising way to expand reasoning scope, whereas we find that simply truncating the max token length can also change its reasoning resilience. Deng et al. (2025); Cheng et al. (2025); Cui et al. (2025) find that entropy impacts the exploration of RL and caps the performance, whereas we introduce the technique to expand entropy.

**RLVR Data Construction.** Few studies examine how to construct RL training prompts to enhance RLVR or RLHF. Gao et al. (2025) propose a principled data-selection method for DPO, showing that overly difficult examples hinder alignment and should be filtered during training. Li et al. (2025a) introduce a strategic selection procedure that identifies key prompts from a full set, achieving comparable RLHF performance with only a subset of the data. Shen et al. (2025) propose Pre-PPO, a data-selection algorithm that enables RL scaling in RLHF by selecting suitable ex-

amples from large datasets. However, RL data construction for RLVR, especially for competitive-programming code generation, remains largely unexplored. To our knowledge, this is the first study to systematically investigate this problem in RLVR, and we analyze both performance and scaling trends on larger models.

**RLVR performance scaling analysis.** Although recent works—e.g., DeepSeek R1 (Guo et al., 2025), SEED-1.5-Thinking (Seed et al., 2025), and Qwen3 (Yang et al., 2025)—present RLVR strategies across multiple benchmarks, few report performance on both small and large models; notable exceptions include DeepSeek R1 and Qwen3. In the RLHF domain, Shen et al. (2025) provide a comprehensive analysis of scaling trends. However, most RLVR evaluations focus on dense 7–32B models, leaving open questions about scaling to much larger models. In this paper, we conduct a comprehensive performance and strategy ablation on Qwen2.5-32B and then directly apply the resulting recipe to on an internal large-scale MoE model,

| Method | LeetCode Pass@10 / Avg@1 | | Codeforces Pass@10 / Avg@1 | |
|---|---|---|---|---|
| SFT Model | 96.88 | / 57.81 | 24.24 | / 11.52 |
| DeepseekV3.1 | 96.88 | / 68.75 | 33.33 | / 16.06 |
| Seed1.6-0715 | 96.88 | / 74.38 | 39.39 | / 18.79 |

*Table 1.* The performance of our SFT model and baselines on Leet-Code Weekly (32 problems) and Codeforces (33 problems, harder).

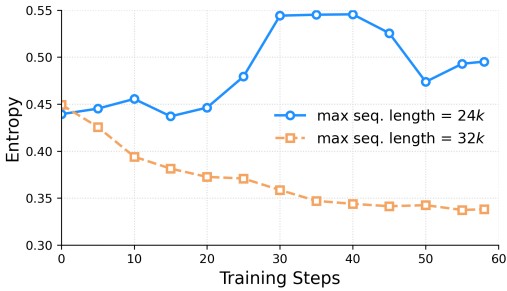

*Figure 3.* The entropy of token selection probability during Stage 1 when setting max sequence length to $24k$ and $32k$.

achieving strong results. This demonstrates that our strategy is effective and suitable for scaling up.

# 3. Method

We present our training pipeline in Figure 2, which consists of two phases, a supervised fine-tuning (SFT) phase that relies on a classifier to upsample hard instances and a two-stage reinforcement learning (RL) phase. We use $k$, $M$, and B to represent $10^3$, $10^6$, and $10^9$ respectively.

## 3.1. Supervised Fine-Tuning

In supervised fine-tuning, we train the base model based on the data distilled from strong models with two additional data curation stages that prioritize the hard samples. Specifically, for the data collection, we collect a set of competitive prompt instances from multiple open-source datasets covering not only programming contest problems but also practical coding such as debugging, querying database, building webpage, etc. We further filter out the responses with incorrect answers. For the first stage of data curation, we select the hard samples using an iterative curriculum-learning approach similar to the arena learning strategy (Luo et al., 2024). We first split the dataset into five folds, and then iteratively train the model on one fold, use the model to infer on the next fold, and only retain the prompts whose inferred answers are incorrect for the subsequent curriculum. In our experiment, this curriculum-learning stage yields a condensed dataset of $470k$ samples from $1.27M$ samples. For the second stage of data curation, we reweight the sam-

ples based on a classifier. The classifier is a 7B model label the problem instances into three categories, easy, medium, and hard. To enhance the coding capability of the model on competitive instances, we double the weight of the instances labeled as hard. Our strategy is similar but simpler than Tong et al. (2024) where the difficulty is labeled according to whether the model can give correct answers. Empirically, we find that the difficulty of the samples, the prompt coverage, and the reasoning traces distilled from strong models are the important factors that contribute to the effectiveness of SFT.

We present the performance of our SFT model in Table 1. The results show that our SFT model achieves comparable performance to strong baselines on moderate tasks, but underperforms them on harder benchmarks, highlighting the role of the subsequent RL phase in improving the performance of our model on hard instances.

## 3.2. Two-Stage Reinforcement Learning

Our motivation for this two-stage RL design is based on the observation that vanilla RL frequently fall into the failure mode: The model is likely to generate similar solutions and thus not able to explore efficiently, indicated by the low entropy of its token selection probability. Also, the model tends to generate repetitive patterns, such as redundant code structures or loops that lead to overly long outputs and trigger truncation (see Appendix A for details). Therefore, the key objective in adopting a two-stage RL is to *expand entropy* in the first stage to make the model explore more efficiently in the second stage.

**Stage 1: Entropy Expansion.** In this stage, we train the model using RLVR on a mixed dataset containing $9k$ coding problems. We collect 8 rollouts for each prompt, and set the max sequence length (including both prompt and response) to $24k$, which is smaller than that used in SFT $32k$. We present the entropy of the token selection probability along with the RLVR training with different max sequence length in Figure 3, and find that limiting the max sequence length to a value less than the SFT phase can mitigate mode collapse and improve exploration. Additionally, we also observe that the repetitive pattern reduces.

**Stage 2: Progressive Curriculum.** In this stage, we train on the most challenging samples in LiveCode V6. Specifically, we train the model under a three-phase progressive curriculum, training the model on top 72, 50, and 25 hardest samples for 64, 32, and 32 gradient steps respectively. We set the max sequence length to $32k$ and collect 80 rollouts per prompt, which is crucial for performance increase. This curriculum design retains the most difficult samples during training and pushes the model to master challenging problems while maintaining performance on easier cases.

| Model | LiveCode 08-11 (166) | LiveCode V5 (167) | LiveCode V6* (175) | LeetCode Weekly (32) | Codeforces Weekly (33) |
|---|---|---|---|---|---|
| QwQ-32B (32k) | 0.578 | 0.569 | 0.537 | 0.472 | 0.124 |
| OpenReasoning-Nemotron-32B (32k) | 0.623 | 0.618 | 0.600 | 0.609 | 0.132 |
| SFT model (32k) | 0.602 | 0.594 | 0.549 | 0.578 | 0.115 |
| RL Stage 1 model (24k) | 0.625 | 0.627 | 0.634 | 0.603 | 0.112 |
| RL Stage 1 + Stage 2 model (32k) | **0.699** | **0.697** | **0.703** | **0.653** | **0.182** |
| **Relative Improvement (RL vs SFT)** | **+16.1%** | **+17.3%** | **+28.1%** | **+13.0%** | **+58.3%** |

*Table 2.* Pass@1 performance comparison across different evaluation benchmarks. The star * indicates that part of LiveCode V6 is used during training. The numbers in parentheses after the benchmark and the model indicate the number of problem instances contained in the benchmark and the max sequence length used in inference respectively.

| Model | LiveCode 08-11 | LiveCode V5 | LiveCode V6 | LeetCode Weekly | Codeforces Weekly |
|---|---|---|---|---|---|
| Basic SFT Strategy | 0.582 | **0.603** | 0.545 | 0.558 | 0.112 |
| W/o Arena Learning | 0.581 | 0.600 | **0.550** | 0.574 | **0.115** |
| W/o Classification | 0.600 | 0.598 | 0.542 | 0.553 | 0.110 |
| Our Strategy | **0.602** | 0.594 | 0.549 | **0.578** | **0.115** |

*Table 3.* Ablation study on different SFT training strategies.

# 4. Experiments

## 4.1. Experimental Setup

- **Models:** The main experiment uses Qwen2.5-32B-Instruct as the base model, and we also use an internal large-scale 406B-A32B MoE model to examine the scaling trend of our method.

- **Data:** For SFT, we collect $1.27M$ prompts from open-source datasets and generate responses by querying from DeepSeekR1-0528. We then condense this dataset to $470k$ prompts using arena learning. For the first stage of RL, we use $9k$ prompts from open source. For the second stage of RL, we use the 175 high-quality samples with comprehensive test cases from LiveCode V6.

- **Experimental Details of SFT:** We train the model for 3 epochs using a learning rate of $1 \times 10^{-5}$ on 256 GPUs with a global batch size 512.

- **Experimental Details of RL:** We use GRPO in the RL training. In the first stage of RL, we train the model for 32 steps and collect 8 rollouts per prompt with the max sequence length $24k$. In the second stage of RL, we train the model for a total of 128 steps and collect 64 rollouts per prompt with the max sequence length $32k$.

- **Evaluations:** We evaluate the models on different benchmarks, LiveCode 08-11 (166 problems), LiveCode V5 (167 problems), LiveCode V6 (175 problems), LeetCode

Weekly (32 problems), and Codeforces Weekly (33 problems). All the problems for evaluation are unseen throughout SFT and RL, except for LiveCode V6 which is used in the second stage of RL. The LeetCode and Codeforces weekly contest problems are released after the collection of our training data, and thus can strictly avoid the data leakage risk.

## 4.2. Experimental on 32B Dense Models

We present the experiment results in Table 2. First, we observe that our model significantly outperforms the previous state-of-the-art 32B model OpenReasoning-Nemotron-32B. The performance of our model is even comparable to the performance of larger models such as DeepSeek-V3.1 (see also Table 5), demonstrating that our model strikes a good balance in the parameter-performance trade-off. Second, the comparison between the RL model and the SFT model indicate that our two-stage RL training results in consistent and significant improvement across different benchmarks, which indicates the effectiveness of our method in effectively boosting the coding capability especially for hard problem instances (see the significant improvement for Codeforces Weekly).

## 4.3. Ablation on SFT

To assess how different SFT strategies affect model generalization, we conduct an ablation comparing four different strategies, while keeping the overall training computation

| Model | LiveCode 08-11 (166) | LiveCode V5 (167) | LiveCode V6[*] (175) | LeetCode Weekly (32) | Codeforces Weekly (33) |
|---|---|---|---|---|---|
| SFT Model | 0.602 | 0.594 | 0.549 | 0.578 | 0.115 |
| RL Only Stage 1 (24$k$) | 0.625 | 0.627 | 0.634 | 0.603 | 0.112 |
| RL Only Stage 1 (32$k$) | 0.676 | 0.688 | 0.675 | 0.592 | 0.102 |
| RL Only Stage 2 (24$k$) | 0.476 | 0.512 | 0.582 | 0.296 | 0.105 |
| RL Only Stage 2 (32$k$) | 0.636 | 0.626 | 0.691 | 0.550 | 0.142 |
| Ours: Stage 1 (24$k$) + Stage 2 (32$k$) | **0.699** | **0.697** | **0.703** | **0.653** | **0.182** |
| Ours: Harder in Stage 2 | **0.712** | **0.707** | **0.743** | **0.678** | **0.188** |

*Table 4.* Ablation study on different RL training strategies. The star [*] indicates that part of LiveCode V6 is used during training.

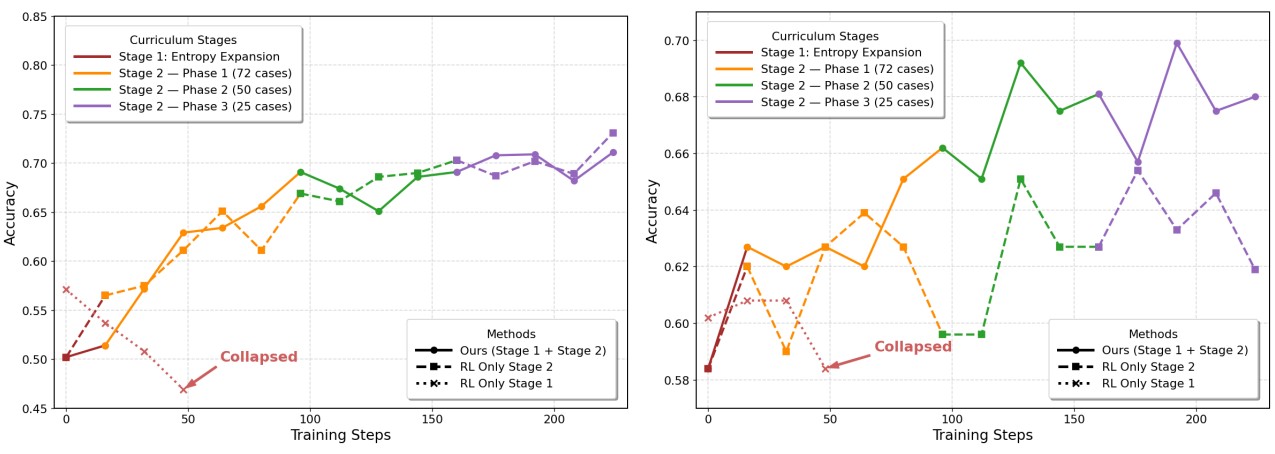

(a) The performance on the training set (LiveCode V6).  (b) The performance on the evaluation set (LiveCode 08-11).

*Figure 4.* The performance of different RL training strategies on the training set of the stage 2 (LiveCode V6) and the evaluation set (LiveCode 08-11) during RL training.

unchanged. In **basic SFT strategy**, we train the base model on the full dataset of 1.27$M$ samples. In **w/o arena learning**, we apply the classification-based reweighting on the all 1.27$M$ before SFT. In **w/o classification**, we train the base model directly on the 470$k$ hard prompts filtered by the arena learning strategy. In **our strategy**, based on the 470$k$ hard prompts selected by arena learning, we use the classifier to label the difficulty and double the weight of hard samples during SFT.

As shown in Table 3, the arena learning strategy maintains performance comparable to the basic SFT baseline (W/o Classification) while reducing the training data by over 60% (from 1.27$M$ to 470$k$ prompts). This highlights the superior efficiency of training on hard samples compared to the full set. Building on this, upweighting hard samples within the reduced dataset leads to further gains. Similarly, applying hard-sample upweighting to the basic SFT strategy improves results on Leetcode V6, LeetCode Weekly, and the Codeforces Weekly Testset, indicating that this approach specifically boosts robustness on harder tasks.

### 4.4. Ablation on RL

To validate the effectiveness of our two-stage RL approach, we compare our method with the following variants: 1) **RL Only Stage 1**, where we conduct RLVR on the 9$k$ training samples as the same in our first stage; 2) **RL only Stage 2**, where we conduct RLVR on the 175 LiveCode V6 samples as the same in our second stage; 3) **Ours: Harder in Stage 2**, where we use 109 curated hard samples from internal data source instead of LiveCode V6. In our proposed method, we set the max sequence length to 24$k$ in the first stage to expand entropy and encourage exploration, and set it to 32$k$ in the second stage to improve performance on hard problems. To validate this design, we try both values for the max sequence length in these ablated variants.

We present the results in Table 4 and plot the performance of different variants on the training set (LiveCode V6) and the evaluation set (LiveCode 08-11) during the RL training in Figure 4. First, we observe that training only on the hard samples in stage 2 leads to stable improvement, whereas training on the samples of mixed difficulty results in performance collapse. This demonstrates the importance of

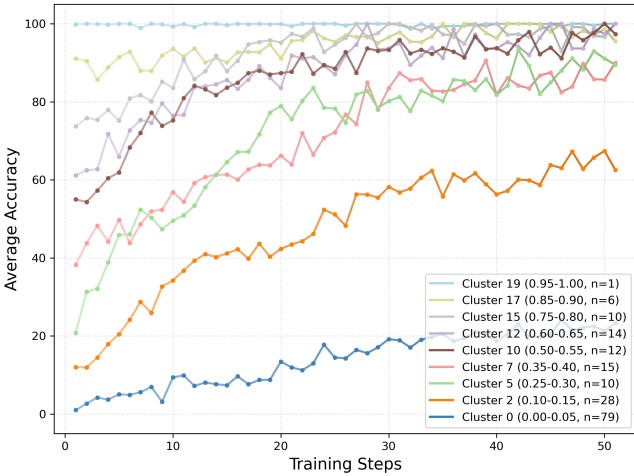

*Figure 5.* The average accuracy of different samples clustered by their initial accuracy through multiple rollouts during RL training. The x-axis indicate the gradient steps during RL training and the legend presents the range of their initial accuracy and the number of samples in the cluster. We randomly select 9 out of 20 clusters for clear view.

training on hard samples instead of mixed samples during RL. Second, the performance can further improve when we use harder (but less) samples in stage 2. This further highlights the importance of RL training on hard samples. Third, RL with only stage 2 (w/o entropy expansion) improves the performance on in-distribution problems (LiveCode V6) but degrades on out-of-distribution benchmarks. This suggests that stage 1 (entropy expansion) helps the model generalize beyond the training distribution. At last, our method (entropy expansion followed by progressive curriculum) achieves better performance than adopting one stage alone, which verifies our design that first expand entropy for exploration by learning on shorter responses and then use large number of rollouts per prompt to explore more on hard samples.

### 4.5. Analysis on RL Training

We also provide in-depth analysis to discover the details of the RL training process.

**How does the model perform on samples of different difficulties during RL training?** To investigate this, we first cluster the samples based on their accuracy in the initial rollouts using the model before RL, and then monitor their accuracy during a canonical RL training. We present the result in Figure 5. We observe that the samples with high initial accuracy does not have much space to improve, and the samples with medium initial accuracy improve significantly during canonical RL training. The key problem is that the canonical RL training does not effectively improve the performance on hard samples, which limits its perfor-

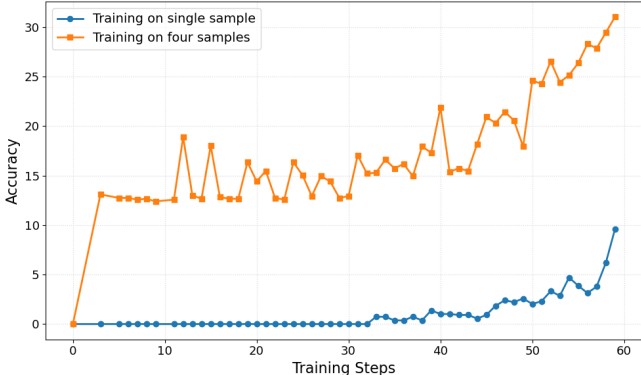

*Figure 6.* The performance on a single hard sample when training only on itself and training on four samples including it.

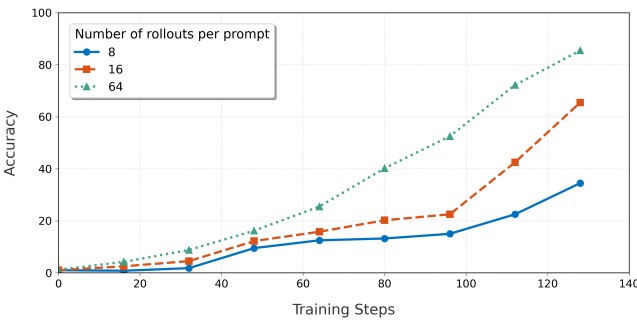

*Figure 7.* Effect of Rollout Number on Single-Case Training Performance.

mance on challenging competitive problems and complex problem-solving tasks. This observation motivates our design that redistribute the compute to the training on hard samples for more efficient RL training.

**Can RL training on hard samples generalize to other samples?** To answer this question, we observe the accuracy of a fixed hard sample during training in two different training schemes, 1) the model is trained only on this single sample, and 2) the model is trained on four samples including this one. We present the result in Figure 6 and find that training only on the single sample performs not as good as training on a mixture of other hard samples. This indicates positive transfer between different training samples during RL training.

**Is the number of rollouts per prompt the larger the better?** Previously in stage 2, we use a larger-than-usual number of rollouts per prompt for better explore on hard samples. While choosing this hyperparameter is a balance of performance and computational costs, we investigate whether increasing it can further improve the performance regardless of the computational cost. To answer this question, we increase this number from 8 to 16 and 64 in the setting where the model is train on only one sample (same

| Model | LiveCode 08-11 (166) | LiveCode V5 (167) | LiveCode V6* (175) | LeetCode Weekly (32) | Codeforces Weekly (33) |
|---|---|---|---|---|---|
| DeepseekV3.1 | 0.692 | 0.713 | 0.693 | 0.688 | 0.161 |
| Seed1.6-0715 | **0.803** | **0.824** | **0.770** | **0.743** | 0.188 |
| Qwen3-2507 | 0.681 | 0.713 | 0.646 | 0.688 | **0.200** |
| SFT Model | 0.681 | 0.692 | 0.656 | 0.627 | 0.155 |
| RL Stage 1 | 0.690 | 0.740 | 0.665 | 0.611 | 0.123 |
| RL Stage 2 | 0.708 | 0.744 | 0.737 | 0.722 | 0.194 |
| **Stage 1 vs SFT** | **+1.32%** | **+6.94%** | **+1.37%** | **-2.55%** | **-20.65%** |
| **Stage 2 vs SFT** | **+3.96%** | **+7.51%** | **+12.35%** | **+15.17%** | **+25.16%** |

*Table 5.* The performance comparison when extending our pipeline to large MoE models. The star * indicates that part of LiveCode V6 is used during training. We set the max sequence length to $64k$ for all models during evaluation.

as the previous setting). We present the result in Figure 7. We find that, increasing the number of rollouts per prompt can lead to steady improvement. Additionally, we observe that training on a single case has negligible impact on the performance of easy samples, but can generalize to improve the performance of other hard samples – further supporting our design that emphasize on learning on hard samples.

### 4.6. Scaling to Large MoE Models

To evaluate the scalability and effectiveness of our training methodology on large-scale mixture-of-experts (MoE) models, we apply our method to an internal MoE model. To customize the training for a larger model, we reconfigure the RL training as follows: In stage 1, we still use the small max sequence length of $24k$, but increase the number of rollouts per prompt to 16. In stage 2, we still use the large number of rollouts per prompt 64, but decrease the number of training steps to 50 (with 20, 20, and 10 steps for the three phases respectively).

We present the result in Table 5 with comparison with other commercial models for reference. First, we observe that after RL stage 1, although the model improves on LiveCode benchmarks, it degrades slightly on more challenging tasks such as LeetCode and Codeforces weekly. This suggest that while stage 1 encourages exploration, a subsequence stage is needed to improve the performance on hard samples. Second, we observe that through the full pipeline of RL (stage 1 + stage 2), the model improves substantially on all benchmarks, especially on the two challenging benchmarks. These results demonstrate the effectiveness of our training pipeline and our data scheduling principle also applies to large MoE models. We also note that, although these experiments are constrained by computational resources and training does not fully converge, it achieves comparable performance to those commercial models.

## 5. Conclusion

In this work, we present a reinforcement learning from verifiable rewards (RLVR) pipeline that focuses on the data scheduling in both the SFT and RL training processes. The pipeline consists of hardness-prioritized supervised fine-tuning (SFT) and two-stage reinforcement learning (RL), which an emphasis on the importance of training on hard samples and entropy expansion for exploration. Our approach achieves state-of-the-art performance among 32B parameter models on competitive code generation, demonstrating that careful data curation and curriculum design are as crucial as algorithmic innovations for the success of RLVR. Moreover, we scale our approach to an internal large-scale MoE model, achieving the performance comparable to other commercial large models, which validates that these data scheduling designs can transfer to a larger scale. While our approach requires more computational resources due to the large number of rollouts per prompt used for exploration, it provides a practical road map to push the model to excellence at challenging problems. Exploring data curation schemes, adaptive curriculum strategies, and efficient rollout techniques to further improve the performance-compute tradeoff in RLVR could be promising future directions.

## Impact Statement

This paper presents work whose goal is to advance the field of Machine Learning. There are many potential societal consequences of our work, none which we feel must be specifically highlighted here.

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

# A. Case Study for Repetition Patterns

Training models directly on algorithmic tasks using max sequence length $32k$ instead of $24k$ proposed in our pipeline often results in significant repetition and inefficient reasoning. In this case study—a **graph isomorphism optimization problem**—the model exhibits various forms of computational redundancy. Note that, given the excessive length of the RL model's raw response, we report only the analysis conducted by GPT-5 regarding these repetitive patterns.

- **Computational Template Repetition**: The core permutation validation algorithm (inverse mapping construction, edge comparison, cost calculation) is applied systematically across dozens of different permutations, each following identical logical steps with different parameters.

- **Granular Analysis Loops**: Each permutation analysis involves repetitive edge-by-edge evaluation following the same pattern: determine desired state → check current state → calculate cost difference → accumulate total, creating highly formulaic micro-computations.

- **Hypothesis-Testing Cycles**: The reasoning exhibits recurring cycles of "try permutation X → calculate cost → compare to expected → express confusion → try next permutation", suggesting systematic but potentially inefficient exploration strategies.

- **Input Processing Redundancy**: Nearly identical parsing and data structure construction logic appears multiple times across different sample inputs, indicating missed abstraction opportunities in edge normalization and set construction.

- **Debugging Iteration Patterns**: Systematic testing of permutation variations (identity, single swaps, compound swaps) using identical verification procedures creates computational redundancy despite thorough coverage.

- **Meta-Cognitive Loops**: Higher-level reasoning patterns repeat the cycle of "acknowledging complexity → attempting enumeration → encountering discrepancies → expressing uncertainty → restarting with variations", suggesting cognitive inefficiencies when facing algorithmic uncertainty.

These patterns suggest that without proper training on diverse reasoning examples within $24k$ context, models tend to fall into repetitive computational habits that, while methodical, lack the efficiency and abstraction capabilities demonstrated by more extensively trained systems.

```
 1  Looking at the content in `<think></think>`, I found extensive repetitive permutation
        calculation patterns. Here are the main repetitive patterns:
 2
 3  ## Repetition Pattern 1: Basic Structure of Permutation Verification
 4
 5  ```python
 6  # This pattern repeats dozens of times in <think>
 7  def verify_permutation_cost(p, n, G_set, H_set, cost):
 8      """Verify the cost calculation for permutation p"""
 9      q = [0] * n
10      for i in range(n):
11          q[p[i]] = i
12
13      total_cost = 0
14      for i in range(n):
15          for j in range(i+1, n):
16              u = min(q[i], q[j])
17              v = max(q[i], q[j])
18              desired = 1 if (u, v) in G_set else 0
19              current = 1 if (i, j) in H_set else 0
20              if desired != current:
21                  total_cost += cost[i][j]
22
23      return total_cost
24  ```
25
26  ## Repetition Pattern 2: Edge State Checking Logic
27
```

```python
28  ```python
29  # Edge state checking that appears repeatedly in <think>
30  def check_edge_state(i, j, q, G_set, H_set):
31      """Check the desired and current state of edge (i,j)"""
32      u = min(q[i], q[j])
33      v = max(q[i], q[j])
34      desired = 1 if (u, v) in G_set else 0
35      current = 1 if (i, j) in H_set else 0
36      return desired, current
37  ```
```

## Repetition Pattern 3: Detailed Permutation Calculations

There are many such repetitive calculations in `<think>`:

```python
# Example extraction of repetitive patterns
def detailed_permutation_analysis():
    """This detailed analysis repeats many times in think"""
    examples = [
        # p = [0, 1, 2, 3, 4] (identity)
        {
            "permutation": [0, 1, 2, 3, 4],
            "analysis": [
                "(0,1): desired=1, current=1 -> 0",
                "(0,2): desired=0, current=1 -> flip: cost[0][2]=1",
                "(0,3): desired=0, current=1 -> flip: cost[0][3]=4",
                "(0,4): desired=0, current=1 -> flip: cost[0][4]=1",
                "(1,2): desired=1, current=0 -> flip: cost[1][2]=5",
                # ... more edge analyses
            ],
            "total": "1+4+1+5+6+3 = 20"
        },

        # p = [0, 1, 2, 4, 3] (swap 3 and 4)
        {
            "permutation": [0, 1, 2, 4, 3],
            "analysis": [
                "(0,1): 0",
                "(0,2): desired=0, current=1 -> flip: cost[0][2]=1",
                "(0,3): desired=0, current=1 -> flip: cost[0][3]=4",
                # ... similar repetitive analysis
            ],
            "total": "some calculation result"
        }
        # There are dozens of such examples in think
    ]
    return examples
```

## Repetition Pattern 4: Sample Verification Structure

```python
# Repetitive pattern for sample input processing
def process_sample_input(sample_name, n, G_edges, H_edges, cost_matrix,
        expected_output):
    """Common pattern for processing sample inputs"""
    print(f"Sample {sample_name}:")
    print(f"N={n}")
    print(f"G edges: {G_edges}")
    print(f"H edges: {H_edges}")
    print(f"Cost matrix: {cost_matrix}")
    print(f"Expected: {expected_output}")

    # Convert to sets
```

```python
 92         G_set = build_edge_set(G_edges)
 93         H_set = build_edge_set(H_edges)
 94
 95         # Try various permutations
 96         min_cost = float('inf')
 97         for perm in all_permutations(n):
 98             cost = verify_permutation_cost(perm, n, G_set, H_set, cost_matrix)
 99             min_cost = min(min_cost, cost)
100
101         print(f"Computed: {min_cost}")
102         return min_cost == expected_output
```

## Repetition Pattern 5: Confusion Expression Statements

```python
# Confusion and retry statements that appear repeatedly in think
confusion_patterns = [
    "I am not finding 9.",
    "But the sample output is 9.",
    "How do we get 9?",
    "This is not matching the sample operations.",
    "Given the complexity...",
    "How about we try the permutation: p = [...]",
    "total = ... -> Not X.",
    "But wait, ...",
    "I see the mistake: ...",
    "Therefore, ..."
]
```

These repetitive patterns indicate extensive trial-and-error processes and redundant calculations in the `<think>` section, which could be simplified by extracting common functions and establishing systematic verification workflows.

