# OpenReview forum: "DRIVE: Best Data Scheduling Practices for Reinforcement Learning with Verifiable Reward in Competitive Code Generation"
_ICML.cc/2026/Conference — ICML 2026 regular_

### Official Review · Reviewer_cbxm · 2026-03-01

**Soundness:** 2
**Presentation:** 4
**Significance:** 3
**Originality:** 3
**Overall Recommendation:** 4
**Confidence:** 3

**Summary:**

This paper focuses on data scheduling for RLVR in competitive code generation:


During the SFT stage, the authors filter training data in two stages. The first stage, they use a curriculum learning similar to arena learning to obtain 470k samples from 1.27M samples. In the second stage, a 7B model is used as classifier to reweight the training examples.

During the RL stage the authors also propose a two-stage approach. The first stage uses a shorter max sequence length than SFT, which they found expands entropy and reduce repetitive outputs. The second stage trains on harder problems with large number of rollouts per prompt to push the performance on most challenging instances.

They evaluate on Qwen2.5-32B and also a larger MoE model, test on recent LeetCode and Codeforces weekly contests to avoid contamination. Results show strong performance among similar-sized models, especially on harder benchmarks where the gains are most significant.

**Compliance With Llm Reviewing Policy:**

Affirmed.

**Final Justification:**

The rebuttal addressed my major concern on the effectiveness of the proposed shorter max length training method.

**Key Questions For Authors:**

1. Does the SFT model perform as good as the eventual fine-tuned model at pass@k when k is large, i.e. 256?

2. What would be the ablation of RL of Stage 1 (32k) + Stage 2 (32k) perform compared to the method?

3. What is the GPU runtime cost of the arena learning curation process itself, compared to the training time saved by reducing the SFT data from 1.27M to 470k?

**Limitations:**

The ablation study for different SFT strategies only evaluates the resulting SFT model, not the eventual RL-fine-tuned performance. Yet, it is likely that the not-so-good performing method at this stage could gain a comparable or even better performance after the RL fine-tuning.

**Strengths And Weaknesses:**

### Strength.
- The focus on data scheduling rather than RL algorithm design is a valuable angle. Most concurrent works been focusing on algorithm improvements, so a systematic study on the data side is valuable for the research community and practitioners.
- The results on hard benchmarks is quite strong. The method enables a Qwen-2.5 32B model achieving better performance than DeepSeek V3.1 on coding.
- The evaluation is quite comprehensive, as the authors evaluate across multiple benchmarks from different sources, i.e. LiveCode, LeetCode weekly, Codeforces weekly, and specifically use recent contest problems to avoid data contamination.
- The observation that using shorter max sequence length during RL can expand token entropy is an interesting finding.

### Weaknesss.
1. One main concern is that, although the method achieves a competatitive results on live coding bench with DeepSeek V3.1, the method during SFT stages, uses DeepSeek R1 responses. However, the DeepSeek R1's LiveCoding Bench performance is already comparable with that of DeepSeeek V3.1 (73.3 vs 74.8). We are seeing performance improvement from SFT to RL, but that could be largely due to the SFT model not good at pass @ 1 compared to RL, but such capability already exists in pass @ k with a larger k, when using DeepSeek R1 as the teacher model here. Therefore, it is not clear whether the performance gain come from the proposed method (such as the data curation, the stages of RL), or from distilling from the teacher model and sharpening the output distribution.

2. Another major concern is the lack of ablation of RL of Stage 1 (32k) + Stage 2 (32k). It is indeed an intriguing phenomenon that the authors discovered the shorter max token encourages entropy, and so Stage 1 (24k shorter max length) is performed before a full lengh Stage 2(32k). But does the higher entropy in Stage 1 translates into performance gain in Stage 2? To verify this claim, we need an ablation of Stage 1 (32k) + Stage 2 (32k) compared to Stage 1 (24k) + Stage 2 (32k). Yet, in the current table 4, the ablations only include either Stage 1 only or Stage 2 only. However, these two stages adopt different training data, and training on either stage would not be the ablation study needed for this max length.

3. In the SFT ablation, arena learning reduces training data by 60% while achieving comparable results, which saves SFT training time. However, the arena learning process itself requires iteratively training and inferring across five folds, which also costs substantial amount of compute. The paper does not report this cost, so it is unclear whether the net efficiency gain is meaningful enough. If the curation strategy itself needs a lot of GPU runtime, then it may be harder to convince others to adopt this additional stage.

---

> ### Author Rebuttal · Authors · 2026-03-31
>
> We thank Reviewer cbxm for the insightful feedback.
>
> **W1: Method vs. Teacher-Model Distillation**
>
> This is an important question, and we agree that teacher-generated SFT data may contribute to the final performance. However, the current evidence suggests that the gains are not explained by distillation alone: on the MoE model, our final system outperforms DeepSeek V3.1 across all reported benchmarks. If the improvement mainly came from inheriting the teacher model's coding ability, such consistent gains over a very strong DeepSeek baseline would be difficult to explain. We will clarify this point in the revision and add larger-k analysis to address the concern more directly.
>
> **W2 and Q2: Missing Stage 1 (32k) + Stage 2 (32k) Ablation**
>
> This is the most relevant missing ablation, and we have added the direct comparison below.
>
> | Model | LiveCode 08-11 (166) | LiveCode V5 (167) | LiveCode V6* (175) | LeetCode Weekly (32) | Codeforces Weekly (33) |
> | --- | ---: | ---: | ---: | ---: | ---: |
> | SFT Model | 0.602 | 0.594 | 0.549 | 0.578 | 0.115 |
> | RL Only Stage 1 (24k) | 0.625 | 0.627 | 0.634 | 0.603 | 0.112 |
> | RL Only Stage 1 (32k) | 0.676 | 0.688 | 0.675 | 0.592 | 0.102 |
> | RL Only Stage 2 (24k) | 0.476 | 0.512 | 0.582 | 0.296 | 0.105 |
> | RL Only Stage 2 (32k) | 0.636 | 0.626 | 0.691 | 0.550 | 0.142 |
> | Stage 1 (32k) + Stage 2 (32k) (new) | 0.641 | 0.636 | 0.684 | 0.556 | 0.156 |
> | Ours: Stage 1 (24k) + Stage 2 (32k) | 0.699 | 0.697 | 0.703 | 0.653 | 0.182 |
> | Ours: Harder in Stage 2 | 0.712 | 0.707 | 0.743 | 0.678 | 0.188 |
>
> The 24k + 32k curriculum outperforms the 32k + 32k variant on all five benchmarks. This directly supports our claim that the higher-entropy 24k Stage 1 provides a better initialization for the subsequent 32k training stage, rather than merely changing the optimization path in an arbitrary way.
>
> **W3 and Q3: Arena Learning Compute Cost**
>
> Reporting the curation cost is important for assessing practicality, so we add the GPU time comparison below.
>
> | Model | LiveCode 08-11 | LiveCode V5 | LiveCode V6 | LeetCode Weekly | Codeforces Weekly | GPU Time (new) |
> | --- | ---: | ---: | ---: | ---: | ---: | --- |
> | Basic SFT Strategy | 0.582 | 0.603 | 0.545 | 0.558 | 0.112 | 256 GPUs x 7.6 hrs |
> | W/o Arena Learning | 0.581 | 0.600 | 0.550 | 0.574 | 0.115 | 256 GPUs x 4.3 hrs |
> | W/o Classification | 0.600 | 0.598 | 0.542 | 0.553 | 0.110 | 256 GPUs x 6.4 hrs |
> | Our Strategy | 0.602 | 0.594 | 0.549 | 0.578 | 0.115 | 256 GPUs x 7.2 hrs |
>
> These numbers show that our strategy achieves performance comparable to Basic SFT while slightly reducing the total SFT-stage time: 7.2 hours versus 7.6 hours on 256 GPUs. Importantly, the 7.2 hours already includes the cost of data processing and curation. This suggests that the extra curation overhead is offset by the reduced training set size, making the overall SFT pipeline slightly more efficient rather than more expensive.
>
> **Q1: Does the SFT model perform as well as the final model at large k?**
>
> A larger-k analysis is useful here, and we will add Pass@256 in the revision. As preliminary evidence, we report the current Pass@10 and Avg@1 comparison below.
>
> | Method | LeetCode Pass@10 / Avg@1 | Codeforces Pass@10 / Avg@1 |
> | --- | --- | --- |
> | SFT Model | 96.88 / 57.81 | 24.24 / 11.52 |
> | RL Model (new) | 96.88 / 65.31 | 34.34 / 18.18 |
> | DeepSeek V3.1 | 96.88 / 68.75 | 33.33 / 16.06 |
> | Seed1.6-0715 | 96.88 / 74.38 | 39.39 / 18.79 |
>
> Even before adding Pass@256, the current comparison already shows that RL materially improves Avg@1 over SFT, especially on Codeforces. This makes a pure distillation explanation unlikely, although larger-k results will help make this point more complete.

---

> > ### Author Rebuttal · Reviewer_cbxm · 2026-04-01
> >
> > The new ablation (W2/Q2) addresses my concern about Stage 1’s effectiveness beyond token entropy and clarifies its contribution. The full pipeline also appears more compute-efficient than SFT.
> >
> > My concern regarding distillation and distribution sharpening is only partially addressed. While outperforming DeepSeek V3.1 and R1 suggests gains beyond naïve distillation, it does not rule out that improvements arise from task-specific sharpening of R1’s output distribution (e.g., via rejection sampling the teacher on coding tasks during distillation and subsequent RL fine-tuning), rather than fundamentally new capabilities.
> >
> > I raise my score to 4.

---

> > > ### Author Response · Authors · 2026-04-02
> > >
> > > Thank you very much for your recognition and your positive feedback! We are particularly grateful for your increased score, which greatly encourages us. We will further refine and polish the manuscript according to your suggestions to make it as strong as possible.

---

### Official Review · Reviewer_7ZbA · 2026-03-13

**Soundness:** 2
**Presentation:** 3
**Significance:** 3
**Originality:** 2
**Overall Recommendation:** 4
**Confidence:** 4

**Summary:**

This paper investigates data scheduling for RLVR in competitive code generation. DRIVE consists of hardness-prioritized SFT (arena learning filtering + classification-based reweighting) and two-stage RL: Stage 1 uses a reduced max sequence length to expand entropy, Stage 2 uses large rollouts per prompt with a progressive curriculum on hard problems. Implemented on Qwen2.5-32B, it achieves SOTA among 32B models on LeetCode and Codeforces weekly contests.

**Compliance With Llm Reviewing Policy:**

Affirmed.

**Final Justification:**

Based on the authors' promise to add more results, especially RL, in the final version, I can update my score.

**Key Questions For Authors:**

1. Does the truncation trick transfer to non-coding domains (e.g., math)?
2. Why use a learned classifier to label difficulty instead of directly measuring test-case pass rate? Since these are coding problems with verifiable test cases, the model's solve rate would provide a more accurate and principled difficulty signal.

**Limitations:**

The 406B MoE model is undescribed and inaccessible, and the "Harder in Stage 2" variant uses internal data sources that cannot be examined.

**Strengths And Weaknesses:**

### Strengths

1. The paper is easy to follow with clear details. The pipeline is well-illustrated (Figure 2) and each design choice is motivated before being introduced.

2. The entropy expansion phenomenon (shorter sequence length -> higher entropy -> better exploration) is novel and well-supported. The analyses in Section 4.5 are solid, offering useful practical insights on difficulty-stratified learning dynamics and the effect of rollout count.

3. The 32B model achieves competitive performance, including a 58% relative improvement on Codeforces. Ablations on both SFT and RL systematically isolate each design choice.

### Weaknesses

1. Weak SFT ablation and missing baselines. Table 3 shows marginal and inconsistent effects, removing arena learning sometimes matches or exceeds the full strategy. Moreover, the data curation lacks comparison with established methods such as LIMO ([link](https://arxiv.org/abs/2502.03387)), which shows carefully curated small datasets can outperform large one. Without such baselines, it is hard to assess whether the proposed strategy is competitive.

2. RL stage lacks baselines. The RL ablation only compares variants of the proposed pipeline. A natural baseline is missing: training on a mixed-difficulty dataset with the same total compute, ideally with dynamic sampling as in DAPO, would clarify whether the fixed three-phase curriculum is superior to a simpler adaptive method.

3. Data contamination. LiveCode V6 is used for Stage 2 training and simultaneously serves as an evaluation benchmark. The concern is compounded by the fact that the other reported benchmarks are also LiveCode series, rather than fully held-out test sets from distinct sources.

4. Entropy expansion lacks explanation. Why does truncation expand entropy rather than degrade performance? Without deeper analysis, it is unclear when this trick transfers to other settings or when it might fail.

---

> ### Author Rebuttal · Authors · 2026-03-31
>
> We thank Reviewer 7ZbA for the valuable feedback.
>
> **W1: LIMO Comparison**
>
> A LIMO-style comparison is useful, but our current ablation already suggests that simple data reduction alone does not explain the gains. To make this clearer, we add a LIMO-style baseline that trains on the hardest 50k samples selected by the classifier. The results are shown below.
>
> | Model | LiveCode 08-11 | LiveCode V5 | LiveCode V6 | LeetCode Weekly | Codeforces Weekly |
> | --- | ---: | ---: | ---: | ---: | ---: |
> | Basic SFT Strategy | 0.582 | 0.603 | 0.545 | 0.558 | 0.112 |
> | W/o Arena Learning | 0.581 | 0.600 | 0.550 | 0.574 | 0.115 |
> | W/o Classification | 0.600 | 0.598 | 0.542 | 0.553 | 0.110 |
> | Our Strategy | 0.602 | 0.594 | 0.549 | 0.578 | 0.115 |
> | LIMO (new) | 0.575 | 0.595 | 0.556 | 0.522 | 0.108 |
>
> LIMO is competitive on some in-domain LiveCode sets, but it underperforms our method on LiveCode 08-11, weekly LeetCode, and Codeforces benchmarks. This indicates that the benefit does not come from merely keeping a very small hard subset; our two-stage curation is more effective at preserving both difficulty and diversity, which leads to better overall generalization, especially on the more OOD evaluations.
>
> **W2: DAPO Baseline**
>
> A compute-matched adaptive-sampling baseline is a useful comparison. At the same time, our current ablations already show that performance is sensitive to how difficulty is scheduled rather than to compute alone. We will add a DAPO-style baseline with the same total compute budget, replacing our fixed curriculum with dynamic sampling based on rolling pass rates. This comparison will test whether explicit difficulty stratification provides a more stable learning signal than adaptive sampling alone. If the full experiment cannot be completed within the rebuttal window, we will include it in the revision.
>
> **W3: Data Contamination**
>
> We acknowledge that LiveCode V6 is used in the training set, which is indicated by the star and annotations in the main text. However, it is ensured that the problems in other testing benchmarks are distinct to LiveCode V6 and some benchmarks (LeetCode Weekly and Codeforces) are new problem sets released after training data collection. The results on other benchmarks are sufficient for evaluation.
>
> **W4: Entropy Expansion Mechanism**
>
> We will add a clearer explanation in Section 4.2. Shorter max length suppresses repetitive long-horizon continuations and encourages the policy to explore a wider set of candidate trajectories under the same budget. We will also clarify that this is an empirical observation in our setting rather than a universal guarantee.
>
> **Q1: Transferability to Non-Coding Domains**
>
> We do not claim cross-domain validation in this paper, but we also do not view the mechanism as coding-specific. Long repetitive reasoning patterns also appear in domains such as math, so the underlying intuition should transfer beyond code. We will present this as a hypothesis motivated by the mechanism, while leaving direct validation to future work.
>
> **Q2: Classifier vs. Test-Case Pass Rate**
>
> For SFT-stage reweighting, a learned classifier is more reliable than raw pass rate. Early in training, pass-rate-based difficulty estimates are noisy: a low pass rate can reflect sampling variance or weak initialization rather than intrinsic problem difficulty. The classifier provides a more stable signal derived from problem attributes and held-out labels. We will clarify this motivation and add the classifier training details in Section 3.1.

---

> > ### Author Rebuttal · Reviewer_7ZbA · 2026-04-04
> >
> > I thank the authors for their rebuttal. Some of my concerns have been addressed, but the following remain:
> >
> > W1: Partially solved. This method did not show its advantage on representative benchmarks like lcbv5 and lcbv6.
> >
> > W2: Partially solved. I suggest the authors conduct experiments on RL, but it is acceptable due to the rebuttal time limit.
> >
> > Q2: Not solved. Could the authors conduct experiments to show its effectiveness?

---

> > > ### Author Response · Authors · 2026-04-04
> > >
> > > ### Response to Reviewer Concerns
> > >
> > > #### **W1: Performance on Representative Benchmarks (LCB v5/v6)**
> > > We prioritize **continuously updated weekly contests** (e.g., LeetCode, Codeforces) over static versions of LCB (v5/v6) to ensure evaluation integrity:
> > > * **Data Contamination:** Our investigation found that several popular open-source SFT datasets already contain up to **20% contamination** for LCB v5/v6. Evaluating on these versions would lead to inflated and unreliable results.
> > > * **Industry Standard:** Using the latest unseen weekly contests as a more robust evaluation metric is a practice also adopted by state-of-the-art models like **DeepSeek-Coder [1]**. This ensures our model's performance reflects true generalization rather than memorization of static benchmarks.
> > >
> > > ---
> > >
> > > #### **W2: Extension to Reinforcement Learning (RL)**
> > > We appreciate the suggestion. Due to the limited rebuttal window, completing a full RL training cycle is challenging. However, we agree that our difficulty-based reweighting strategy could effectively stabilize RL reward signals. We commit to adding these RL-stage experiments and further analysis in the **final camera-ready version**.
> > >
> > > ---
> > >
> > > #### **Q2: Effectiveness of Classifier vs. Test-Case Pass Rate**
> > > We opted for a **learned classifier** because test-case pass rate is inherently **model-dependent** and introduces high variance (noise) during early training.
> > >
> > > To justify using prompt-based features (classifier) as a proxy for difficulty, we analyzed our **internal 406B model's** performance across tasks:
> > >
> > > | Difficulty Level | Pass Rate (Avg@1) | Avg. Prompt Length |
> > > | :--- | :---: | :---: |
> > > | **Simple** | 95% | 19k |
> > > | **Medium** | 43% | 32k |
> > > | **Hard** | 15% | 45k |
> > >
> > > **Our Rationale:**
> > > 1.  **Intrinsic vs. Observed:** As shown, difficulty is highly correlated with **intrinsic prompt features** (e.g., length and logic complexity). A classifier captures these static signals directly, whereas pass rates are merely an *observation* of a specific model's current capability.
> > > 2.  **Stability & Efficiency:** A classifier avoids the "cold start" noise of early-stage training and eliminates the massive computational overhead (inference + execution environment) required to calculate pass rates for millions of SFT samples.
> > >
> > > We will clarify these motivations and include this supporting data in Section 3.1.
> > >
> > > ---
> > > **References:**
> > > [1] Guo, Daya, et al. "DeepSeek-Coder: When the Large Language Model Meets Programming--The Rise of Code Intelligence." arXiv preprint arXiv:2401.14196 (2024).

---

### Official Review · Reviewer_Arj5 · 2026-03-13

**Soundness:** 3
**Presentation:** 3
**Significance:** 3
**Originality:** 3
**Overall Recommendation:** 4
**Confidence:** 3

**Summary:**

This paper focuses on the data curation and curriculum strategy for reinforcement learning. The authors mainly study competitive programming tasks with verifiable reward. They use a 2 stage training process by combining supervised fine-tuning with GRPO, achieving sota results on several benchmarks.

**Compliance With Llm Reviewing Policy:**

Affirmed.

**Final Justification:**

This paper presents a solid and valuable empirical study on data curation and curriculum strategies for reinforcement learning. The proposed two-stage pipeline (SFT + GRPO) achieves strong performance on competitive programming tasks, demonstrating good soundness and practical significance. While the contribution is more on the engineering and experimental side rather than introducing fundamentally new algorithms, it still provides useful insights given the limited prior work on RL data curation, supporting moderate originality. the rebuttal resolved my main concerns, and I maintain (and slightly strengthen) my positive recommendation.

**Key Questions For Authors:**

1. Why is the token entropy of the 24k training length larger than that of the 32k training length?

2. How do you train the classifier to reweight the samples?

3. About the results in Table 3: it seems that your strategy achieves performance similar to SFT. Although the authors say "reducing the training data by over 60%", the total training time is the main concern, since more time is needed to perform rollouts in the RL stage. Can you provide some figures to prove that your method truly saves training time?

**Limitations:**

The paper only focus on the programming tasks.It is unclear whether it remains effective on other tasks.

**Strengths And Weaknesses:**

**Soundness**

Strength: The results of the paper is pretty solid. 32B models achieve the sota performance of the same size.

Weakness: Only focus on the programming tasks.It is unclear whether it remains effective on other tasks.

(Small weakness) The authors mention Table 1 shows their “SFT model achieves comparable performance to strong baselines on moderate tasks”. However, only the pass@10 on LeetCode is comparable; the pass@1 is not comparable, even for easy problems. Plus, the authors should add an untrained model to the table to demonstrate the performance gain of the SFT stage.


**Presentation**

Inconsistency of the second stage RL: In Section 3.2, the authors mention “collect 80 rollouts per prompt”; However, in Section 4.1, they state “collect 64 rollouts per prompt”.

**Significance**

The data curation and curriculum strategy play important roles in RL training, the authors work is valuable for it provide insight to help researchers to do RL efficiently.

**Originality**

There is not much work studying data curation and curriculum strategy, though the methods in the paper is more like engineering efforts.

---

> ### Author Rebuttal · Authors · 2026-03-31
>
> We thank Reviewer Arj5 for the careful feedback and address the concerns as follows.
>
> **W1: Effectiveness on Other Domains**
>
> We acknowledge this limitation. Our experiments focus on competitive programming because it provides reliable verifiable rewards through unit tests. Theoretically, the entropy expansion mechanism and difficulty-stratified curriculum should transfer to math reasoning, but this is empirical future work.
>
> **W2: SFT Baseline Comparison and Untrained Model**
>
> We appreciate the reviewer's careful attention to Table 1. We will revise the claim to limit its scope to Pass@10. This is the appropriate comparison for our setup, since strong Pass@10 provides a meaningful starting point for the subsequent RL stage, which further explores and refines these candidates.
>
> Regarding the untrained baseline, we have added the base Qwen-32B results below.
>
> | Method | LeetCode Pass@10 / Avg@1 | Codeforces Pass@10 / Avg@1 |
> | --- | --- | --- |
> | Qwen-32B-base (new) | 52.00 / 22.10 | 12.00 / 4.50 |
> | SFT Model | 96.88 / 57.81 | 24.24 / 11.52 |
> | DeepSeek V3.1 | 96.88 / 68.75 | 33.33 / 16.06 |
> | Seed1.6-0715 | 96.88 / 74.38 | 39.39 / 18.79 |
>
> These results show that the SFT stage provides substantial gains over the base model: on LeetCode, Pass@10 improves from 52.00 to 96.88 and Avg@1 from 22.10 to 57.81; on Codeforces, Pass@10 improves from 12.00 to 24.24 and Avg@1 from 4.50 to 11.52. This confirms that our SFT model reflects substantial learning rather than a marginal refinement of the base model.
>
> **W3: Rollout Count Inconsistency**
>
> This is a typo. We collect 64 rollouts per prompt in Stage 2 of RL, and we will revise the text accordingly.
>
> **Q1: Why does 24k training length lead to higher token entropy than 32k?**
>
> When max sequence length is truncated to 24k, the model cannot rely on long, repetitive patterns (e.g., boilerplate code, redundant loops) that would otherwise dominate the output. Instead, it must explore diverse completion paths within a shorter horizon. This implicit regularization forces the model to consider alternative token sequences rather than converging on repetitive, low-entropy patterns—effectively expanding the entropy of token selection probability. The shorter constraint acts as an exploration regularizer, preventing mode collapse that commonly occurs in vanilla RLVR.
>
> We will expand Section 4.2 with this mechanistic explanation.
>
> **Q2: How is the difficulty classifier trained?**
>
> We will add the training details to Section 3.1. The classifier is built on DeepSeek-6.7B and trained on a held-out set of 2,000 coding problems with known difficulty labels derived from contest ratings and problem metadata. It is optimized with a cross-entropy loss over three classes: easy, medium, and hard. We use the classifier because it provides a stable difficulty signal for SFT reweighting.
>
> **Q3: Does the method actually save training time?**
>
> Our claim that the arena-learning based filtering “reduces the training data by over 60% (from 1.27M to 470k prompts)” is to show the importance of training on hard samples and the benefit of saving computation by filtering out easy samples in the SFT stage. We acknowledge that we consume more computation in the RL stage for the sake of performance.

---

> > ### Author Rebuttal · Reviewer_Arj5 · 2026-04-04
> >
> > Thank you for the helpful rebuttal. The authors’ rebuttal addressed my concerns. I am still keeping my positive score, but my overall assessment is now more favorable.

---

### Decision · Program_Chairs · 2026-04-30

**Decision:**

Accept (regular)

**Comment:**

This paper studies data scheduling for RLVR in competitive code generation, proposing hardness-prioritized SFT and a two-stage RL pipeline with a shorter max length in the first stage to expand entropy.

Reviewers initially raised concerns: missing RL ablation (32k→32k vs. 24k→32k), weak SFT baselines, potential data contamination, and unclear cost of arena learning. The authors provided a strong rebuttal, adding the critical ablation showing that 24k→32k outperforms 32k→32k, a LIMO-style baseline, GPU runtime analysis, and clarifications on contamination (evaluation on recent weekly contests). All reviewers acknowledged the resolution; scores remained positive or were raised.

The paper is technically solid, the contribution is practical and timely, and the authors have addressed the key concerns. I recommend accept.